# Distribution of Pesticides and Polychlorinated Biphenyls in Food of Animal Origin in Croatia

**DOI:** 10.3390/foods13040528

**Published:** 2024-02-08

**Authors:** Maja Đokić, Tamara Nekić, Ivana Varenina, Ines Varga, Božica Solomun Kolanović, Marija Sedak, Bruno Čalopek, Ivana Kmetič, Teuta Murati, Darija Vratarić, Nina Bilandžić

**Affiliations:** 1Laboratory for Residue Control, Department of Veterinary Public Health, Croatian Veterinary Institute, Savska Cesta 143, 10000 Zagreb, Croatia; dokic@veinst.hr (M.Đ.); tamara.nekic@gmail.com (T.N.); varenina@veinst.hr (I.V.); varga@veinst.hr (I.V.); solomun@veinst.hr (B.S.K.); sedak@veinst.hr (M.S.); calopek@veinst.hr (B.Č.); 2Laboratory for Toxicology, Faculty of Food Technology and Biotechnology, University of Zagreb, Pierottijeva 8, 10000 Zagreb, Croatia; ikmetic@pbf.hr (I.K.); teuta.murati@pbf.unizg.hr (T.M.); 3Veterinary and Food Safety Directorate, Ministry of Agriculture of Republic of Croatia, Planinska 2a, 10000 Zagreb, Croatia; darija.vrataric@mps.hr

**Keywords:** pesticides, polychlorinated biphenyls, fat, meat, GC-MS/MS, multi-residue analysis, food safety, risk assessment

## Abstract

Pesticides and polychlorinated biphenyls (PCBs) are persistent environmental pollutants. When entering the food chain, they can represent a public health problem due to their negative effects on health. In this study, concentrations of organochlorine pesticides (OCPs), organophosphate pesticides (OPPs), pyrethroids, carbamates, and PCBs—a total 73 compounds—were determined in a total of 2268 samples of fat tissues (beef, pork, sheep, goat, poultry, game, horse, rabbit) and processed fat, meat, and processed meat products collected in Croatia during an 8-year period. In fatty tissues, 787 results exceeded the limits of quantification (LOQ): 16 OCPs, eight OPPs, six pyrethroids, one carbamate, and seven PCBs. The most positive results in fat samples were found for OCPs, with a frequency of quantification in the range of 57.5–87.5%. Hexachlorobenzene (HCB) and dichlorodiphenyldichloroethylene (DDE) were quantified in the highest percentages, in the ranges of 5.5–66.7% and 5.4–55.8%. Concentrations above the MRL values were determined for chlorpyrifos in pork fat and for resmethrin in six fat samples and one pâté. In 984 samples of meat and meat products, only 62 results exceeded the LOQ values. The highest frequency of quantification was determined for OCPs (25 samples), of which 40% were DDT isomers (60% DDE). Frequency quantifications of PCBs in fat samples were between 7.23 and 36.7%. An evaluation of the health risk assessment showed that the consumption of fat, meat, and meat products does not pose a threat to consumer health, since all EDI values were well below the respective toxicological reference values.

## 1. Introduction

Pesticides are an indispensable tool for plant protection and increasing crop yields. However, their progressive and increasing production and their use in agriculture often cause unwanted effects in the environment, animals, and people. Pesticides are among the main factors of environmental pollution [1]. More than 1000 of these compounds are included in groups of compounds, such as organochlorines, organophosphates, carbamates, and synthetic pyrethroids, which are designed to be toxic to pests and disease vectors and are marketed as herbicides, insecticides, fungicides, rodenticides, nematicides, etc. The production and use of synthetic organic chemicals, organochlorine pesticides (OCPs), and polychlorinated biphenyls (PCBs) has been banned in many countries since the 1970s and 1980s due to their harmful effects on humans [2]. Due to their persistent nature and slow degradation, these compounds have remained in the soil and in water ecosystems, making them persistent organic pollutants (POPs) [3]. They enter the food chain through soil, sediments, air, and water, bioaccumulating in plants and animal and human tissues. Their concentrations increase significantly as they move up the food chain [4,5,6]. Due to their high lipophilicity, residues are found in the adipose tissue of all animals and humans [3,5,7]. It is also characteristic of these compounds that they are more resistant to biotransformation by organisms and remain in the body longer [8,9].

Due to their physicochemical properties, PCB compounds have numerous applications in industries such as electronics manufacturing and in construction, including transformers, turbines, heat exchange fluids, vacuum pumps, etc. PCBs are incorporated into a number of products and materials such as elastic sealants, paint additives, plastics, adhesives, synthetic resins, rubber, and many others. They are also used as vehicles for pesticides (pesticide extenders) [5]. This wide application results in their permanent environmental presence [3,5].

Since PCBs and OCPs are highly lipophilic, the main source of human exposure to PCBs is via fatty animal products, including fish [5,10]. The general population that often consumes food of animal origin is therefore more exposed [9,11]. The literature indicates that the contribution of OCP compound exposure via meat and meat products, dairy products, and fish and other fruits can exceed 90% [6,12,13]. Long-term exposure to these toxic pollutants through the consumption of food can lead to intoxication and cause a number of health problems that include the liver, skin, reproductive, endocrine, neurological, and immune systems and can lead to the prevalence of chronic diseases and cause the appearance of various cancers [7,9]. The International Agency for Research on Cancer (IARC) has classified certain OCP compounds, i.e., chlordane, heptachlor, hexachlorocyclohexanes (HCH), hexachlorobenzene (HCB), and toxaphene, in group 2B as possible human carcinogens, whereas DDT (4,4′-dichlorodiphenyltrichloroethane) and dieldrin are classified as 2A, that is, as probably carcinogenic to humans. Polychlorinated biphenyls and lindane are classified in group 1 as carcinogenic to humans [14].

Organophosphate pesticides (OPPs) are applied as effective insecticides, acaricides, and miticides and are commonly used to treat stored grains. The most frequently used OPPs are chlorpyrifos, chlorpyrifos-methyl, phorate, dimethoate, diazinon, malathion, acephate, azinphos-methyl, phosmet, and dicrotophos [15]. Symptoms of acute intoxication in humans can include dizziness, nausea, headache, spasms, convulsions, loss of reactions, and even death [15]. OPPs cause neurotoxic disorders in humans, with cholinergic syndrome leading to millions of cases of poisoning annually, with a fatal outcome of more than 15% each year [16]. Also, the possible association between exposure to organophosphorus pesticides and neurodegenerative diseases, attention deficit disorder, dementia, hyperactivity, and Parkinson’s disease is still being investigated [3,16].

Pyrethroids are insecticides that are increasingly used in agriculture and veterinary medicine. They are widely used due to their low cost, effectiveness at low doses, low toxicity for mammals, and high environmental degradation rates in comparison to other classes of pesticides (OCPs, OPPs, and carbamates) [17,18]. Toxicological studies have shown that these compounds have effects such as potential endocrine disruption, carcinogenicity, and neurotoxicity [19,20,21].

Considering the number of negative effects on human health and for the purpose of consumer risk assessment and food safety, international regulatory, advisory, and scientific organisations have focused on creating reliable POP monitoring systems. The Stockholm Convention on Persistent Organic Pollutants (POPs) was adopted and recommends monitoring OCPs and six indicator congeners of PCBs (PCB-28, PCB-52, PCB-101, PCB-138, PCB-153, and PCB-180) that are present in elevated concentrations in the environment, food, and human tissues [5,22]. At the European Union (EU) level, regulations defining the control of pesticides by EU Member States in primary production, i.e., in the processing of products of animal origin, have been adopted and are being improved [23,24,25]. The EU has also prescribed maximum residue level concentrations (MRLs) for the sum of six indicator congeners of PCBs in meat and meat products, fatty tissue, animal offal and milk, milk products, and eggs [26,27]. The maximum permissible levels of pesticide residues (MRL) in food are regulated by Regulation 396/2005/EC [28]. 

Pesticide analysis is performed with robust, reliable, sensitive, and selective techniques of gas chromatography–triple quadrupole mass spectrometry (GC-MS/MS) [29,30] and liquid chromatography–triple quadrupole mass spectrometry (LC-MS/MS) [31,32], which together enable the analysis of hundreds of compounds. The poor performance of GC-MS with respect to polar and ionic compounds of pesticides due to poor thermal stability or volatility replaces LC-MS as a good choice for the analysis of multiple pesticide residues [33]. More recently, gas chromatography (GC) and high-performance liquid chromatography (HPLC) have been used in combination with high-resolution mass spectrometry (HGMS), providing high sensitivity and good accuracy, and representing the most advanced instruments in modern multiresidue pesticide analysis [34].

In this study, the presence and distribution of OCLs, OCPs, pyrethroids, carbamates, and PCBs were investigated in foods of animal origin collected in Croatia during an 8-year period. Furthermore, the goal was to determine daily intake to assess the risk of consumer exposure based on the established mean values of pesticide concentrations.

## 2. Materials and Methods

### 2.1. Sample Collection

In total, 1284 fatty tissues and processed fat, as well as 984 samples of meat and meat products, were collected and analysed during the period of 2015–2022 (Table 1). Meat and fat of different animals were collected from different farms in Croatia as part of the National Residue Monitoring Programme. 

Processed meat products and processed fat were purchased from grocery stores and supermarkets in different Croatian cities. Processed meat products (Table 1) included various products, such as pork products (bacon, ham, prosciutto, sirloin), canned products (beef vegetable stew, pork vegetable stew, beef breakfast meat, pork breakfast meat), pâté products, salami products (truffle salami, mortadella salami, chicken salami), hamburger, and sausages. After delivery to the laboratory, samples were homogenised and stored frozen at −18 °C until analysis.

### 2.2. Chemicals and Reagents

Certified pesticide standards (purity 94–99%) were obtained from Dr. Ehrenstorfer LGC Standards (Augsburg, Germany), Toronto Research Chemicals (Toronto, ON, Canada) and Sigma Aldrich (Seelze, Germany). Individual stock solutions of pesticides at concentrations of 1000 µg/mL were prepared in acetone or dimethylformamide and stored in amber screw-capped glass vials at below −18 °C. Mixed standard solutions for validation and calibration were prepared by appropriate dilutions of stock standard solutions with acetonitrile. LC-MS purity-grade acetonitrile, acetone, cyclohexane, and ethyl acetate were supplied by Honeywell (Charlotte, North Carolina, USA). Dimethylformamide, sodium sulphate (anhydrous), and sodium chloride were obtained from Sigma Aldrich (Bellefonte, Pennsylvania, USA). Tributyl phosphate (Sigma-Aldrich, Seelze, Germany) was used as an internal standard and prepared as a spiking solution at concentrations of 10 µg/mL in acetonitrile. Ultrapure water (18.2 MΩ/cm) was produced by the Direct-Q^®^ 5 UV System (Millipore Corporation Merck, Darmstadt, Germany).

### 2.3. Sample Preparation for Fat Samples

For analysis, 7.5 g fat was weighed into a glass beaker with the addition of 5 g sodium sulphate. To this, 6–7 mL GPC solvent (cyclohexane and ethyl acetate, 1:1, *v*/*v*) was added, and the beaker was warmed over a steam bath. The contents were mixed using a glass rod until the fat was completely dissolved, then removed from the steam bath before the solvent boiled. A small plug of cotton wool was placed in a glass funnel mounted on a measuring cylinder. The cotton was conditioned with 1–2 mL GPC solvent. The supernatant was decanted into the measuring cylinder through the funnel. A further 6 mL GPC solvent was added and repeated. The funnel was rinsed with 1–2 mL GPC solvent and collected in the measuring cylinder in a water bath. After 10 min, the volume was increased up to 25 mL with GPC solvent.

The extract (8 mL) was cleaned by gel permeation chromatograph (GPC, LC-20 Prominence, Shmidazu, Tokyo, Japan) based on the following conditions: mobile phase, cyclohexane + ethyl acetate (1 + 1, *v*/*v*); flow rate, 3 mL/min; detection wavelength, 254 nm; injection volume, 2 mL; commenced collecting time, 26 min; stopped collecting time, 47 min. The eluted portions of 26–47 min were collected in fraction collection tubes and concentrated to 1 mL in a concentrator with a gentle stream of nitrogen (12 ± 2 psi) at a temperature of 35 ± 5 °C.

Then, 1 mL extract was transferred to a Chem Elut cartridge, washed using 2 × 1 mL hexane/acetone solvent, and left to stand for at least 60 min. The Chem Elute cartridge was placed directly above the two silica cartridges. The silica cartridges were conditioned with 7 mL hexane-saturated acetonitrile, and the sample was eluted from the Chem Elute cartridge with 3 × 6 mL portions of hexane-saturated acetonitrile. The eluate was reduced to 1 mL in a concentrator with a gentle stream of nitrogen (12 ± 2 psi) at a temperature of 35 ± 5 °C. The internal standard solution and standard were added to the sample before injection into the GC-MS/MS. Matrix-matched calibration was used for calibration.

### 2.4. Sample Preparation for Meat and Meat Products

For analysis, 15 g meat or meat products was weighed into a centrifuge bottle with the addition of 100 mL hexane/acetone extraction solvent, followed by 20–30 g anhydrous sodium sulphate. The mixture was blended using a vortex blender for approximately 1 min, and the blender blades were rinsed with 3 mL extraction solvent. The extract was centrifuged at 2500–3500 rpm for 2 to 3 min. The supernatant was decanted through the sodium sulphate column and filtered through the sodium sulphate into the tube. In the bottom layer, 60 mL hexane/acetone extraction solvent was added again, and the steps were repeated. The column was rinsed with 20 mL hexane/acetone extraction solvent. The eluted portions were concentrated to 2 mL in a concentrator with a gentle stream of nitrogen (12 ± 2 psi) at a temperature of 35 ± 5 °C. The extract was transferred to a labelled graduated test tube, and the volume increased to 10 mL with GPC solvent. The test tube was cleaned based on the following conditions by GPC: mobile phase, cyclohexane + ethyl acetate (1 + 1, *v*/*v*); flow rate, 3 mL/min; detection wavelength, 254 nm; injection volume, 2 mL; commenced collecting time, 26 min; stopped collecting time, 47 min. 

The eluted portions of 26–47 min were collected in fraction collection tubes and concentrated to 1 mL in a concentrator with a gentle stream of nitrogen (12 ± 2 psi) at a temperature of 35 ± 5 °C. The extract (1 mL) was transferred into a Chem Elut cartridge, washed using 2 × 1 mL hexane/acetone solvent, and left to stand for at least 60 min. The Chem Elute cartridge was placed directly above two silica cartridges. The silica cartridges were conditioned with 7 mL hexane-saturated acetonitrile, and the sample was eluted from the Chem Elute cartridge with 3 × 6 mL portions of hexane-saturated acetonitrile. The eluate was reduced to 1 mL in a concentrator with a gentle stream of nitrogen (12 ± 2 psi) at a temperature of 35 ± 5 °C. The internal standard solution and standard were added to the sample before injection into GC-MS/MS. Matrix-matched calibration was used for calibration.

### 2.5. GC-MS/MS Analysis

The concentrations of the 73 tested compounds (23 OCPs, 29 OPPs, 10 pyrethroids, 4 carbamates, and 7 PCBs) were determined using a GC-MS/MS system equipped with an Agilent gas chromatograph 7890A (Palo Alto, CA, USA), autosampler series 7693B, split/splitless injector in pulsed splitless mode and tandem mass spectrometry detector 7000B with electron impact type ionisation source. Chromatographic separation was performed on a HP-5 MS capillary column (30 m × 0.25 mm ID, 0.25 µm, Agilent Technologies, Santa Clara, California, USA) with helium (99.9999% purity) at a constant flow rate of 0.9 mL/min as the carrier gas. The injection volume was 2 μL. The oven temperature programme was set as follows: initial temperature of 70 °C and hold for 1 min, followed by a 25 °C/min ramp to 150 °C, followed by a 3 °C/min ramp to 200 °C and hold for 5 min, and finally followed by an 8 °C/min ramp to 280 °C and hold for 13 min. Analysis run time was 44.867 min. Other operating conditions were as follows: inlet temperature of 80 °C till 0.01 min, then 720 °C/min to 280 °C; transfer line temperature 280 °C; source temperature 300 °C; temperature MS1 and MS2 quadrupoles 150 °C; collision gas (N_2_) flow 1.5 mL/min; quench gas (He) flow 2.25 mL/min. The GC-MS/MS system was controlled by Mass Hunter software version B.07.01. Quantitative and qualitative analysis was performed with Mass Hunter software based on the two most intensive precursor ion to product ion Multiple Reaction Monitoring (MRM) transitions. The values of the GC-MS/MS optimised parameters for each MRM transition for the respective analytes are presented in the Appendix A.

### 2.6. Method Validation

Validation of the method was performed according to the EU quality control procedures defined in SANCO/12571/2013 [35] and SANTE/11945/2015 [36]. The analytical parameters evaluated were linearity, recovery, and method repeatability. Intra- and inter-day precision and uncertainty were also estimated. The linearity of the method was evaluated by establishing matrix-matched calibration curves in fat and muscle samples. Seven calibration levels of 1, 2, 5, 10, 50, 100, and 200 μg/kg were prepared by spiking the corresponding blank extracts of muscle previously prepared by extraction.

Recoveries and repeatability of the method were studied at four calibration levels by spiking blank matrix meat (2, 5, 10, and 50 μg/kg) and fat (0.5, 1, 2, 5, and 10 μg/kg). Reaching a LOQ of 2 μg/kg is important and challenging, despite the fact that MRLs are usually set in the range of 0.01–1 mg/kg or lower. Intra-day precision was studied by injecting spiked samples for five different days. Inter-day precision was studied by calculating the RSDs obtained from checking the recoveries of five spiked samples on the same day. Uncertainty of the overall procedure was also estimated using validation data.

### 2.7. Estimation of Daily Intakes and Health Risk Assessment

The estimated daily intake (EDI) was calculated to assess the risk of consumer chronic dietary exposure to pesticide residues in meat and meat products and animal fat samples. The EDI (µg/kg bw/day) was calculated using Equation (1): EDI = C × MS,(1)
where C is the pesticide concentration (µg/kg w.w.), MS is the meal size (g per portion of meat, meat products, and fat per day and body weight, g/kg bw/day). Data on food consumption of the adult Croatian population are available on the EFSA Comprehensive European Food Consumption Database [37]. Chronic consumptions for adult consumers expressed as g/kg bw/day were 2.65 of meat and meat products and 0.40 of animal fat. 

The EDI was calculated for the most commonly used types of fat in the diet such as beef, pork and poultry fat, processed fat, and additionally for meat and meat products for which pesticide concentrations and PCBs were quantified more than once (beef, pork, and pâté).

To estimate the risk of carcinogenic and noncarcinogenic effects for all quantified pesticides and POPs, the Hazard Quotient was calculated as the ratio of EDI to the toxicological reference value [38]. The risk to human health was characterised by comparison with the toxicological reference value approved at the European or international level and defined by the Joint FAO/WHO Meeting on Pesticide Residues, EFSA, or by other international bodies and safety agencies such as ANSES (French Agency for Food, Environmental, and Occupational Health and Safety), the United States Environmental Protection Agency (USEPA), and the Agency for Toxic Substances and Disease Registry (ATSDR). The toxicological reference value used for chronic risk assessment is the acceptable daily intake (ADI) for pesticides within the scope of Regulation (EC) 1107/2009, available in the frame of the EU Pesticides database [39]. For pesticides for which ADI values have not been defined, such as aldrin, endrin, heptachlor, or DDT, the provisional tolerable daily intake (PTDI) as defined by WHO Expert Group on Pesticide Residues can be applied, and these values are available in the database of the World Health Organization (WHO) and in reports of the Joint FAO/WHO Meeting on Pesticide Residues (JMPR) [40]. The USEPA has defined an oral reference dose (RfD) of 20 ng/kg/day for PCBs (Aroclor 1254) based on dermal/ocular and immunological effects in monkeys [41]. 

The Hazard Quotient approach was calculated using Equation (2):HQ = EDI/ADI.(2)

No adverse effect is likely to occur (health-protective) when HQ is <1. However, if HQ is >1, a high level of concern is indicated for a chronic effect occurrence, and the higher the HQ, the higher the concern for chronic toxic effects, highlighting the need for immediate risk management actions.

### 2.8. Statistical Analysis

Statistical analyses were performed using Statistica 10 (StatSoft^®^ Inc., Tulsa, Oklahoma, USA) and Excel (Microsoft Excel, 2010). The concentrations of pesticides and PCBs in different meat and meat products and fat were expressed as the mean ± standard deviation (SD) and minimum and maximum. Only results above the LOQ value were statistically analysed. If all obtained results in the group of samples were below the LOQ, the results are presented as <LOQ.

## 3. Results and Discussion

### 3.1. Method Validation

Method validation and quality control for fatty tissues and meat and meat products were conducted following the European Commission SANCO/12571/2013 [35] and SANTE/11945/2015 [36]. The calculated limits of quantification LOQs and other validation metrics for each compound are summarised in Appendix A for fat and Appendix A for meat. The method was validated in terms of the linearity of at least five levels. Deviation of the back-calculated concentration from the true concentration was ≤20%. The limit of quantification (LOQ) was determined as the lowest spike level for which the acceptance criteria according to SANCO/12571/2013 [35] and SANTE/11945/2015 [36] were met. For fatty tissues, LOQ values were 2 and 5 μg/kg, and for meat and meat products, the LOQ values ranged between 0.5 and 10 μg/kg. Recovery values at four spiking levels for meat and products were 80.76–111.14%, and the intra-day and inter-day precisions were less than 20% (3.23–17.90%). Recovery and precision values at five calibration levels for fat ranged from 80.96–117.76%, with an RSD from 4.54–18.90%. Results obtained by validation indicated good accuracy and precision.

### 3.2. Pesticide Residues

Concentrations of 73 compounds (23 OCPs, 29 OPPs, 10 pyrethroids, four carbamates, and seven PCBs) in different animal fat samples and processed fat collected in Croatia during the period of 2015–2022 are shown in Table 2. The results contain only those concentrations exceeding the LOQ values. Concentrations above the LOQ were determined for 16 OCPs, eight OPPs, six pyrethroids, one carbamate, and seven PCBs. Beef, pork, and poultry fat had the highest number of results above the LOQ value. However, the highest frequency of detection of the analysed compounds, in relation to the number of analysed samples, was determined for sheep fat (132.7%), goat fat (118.8%), and game fat (178%). Among the four pesticide groups analysed, the highest number of quantified results was found for OCPs, ranging between 57.5% for pork fat and 87.5% for game fat. Among the total 787 results above the LOQs, eight samples had concentrations above the MRL values (chlorpyrifos in pork fat, resmethrin in six fat samples and one pâté).

However, the frequency of pesticides and PCBs exceeding the LOQ values for meat and meat products was significantly lower than that for fatty tissues (Table 3). Of the 984 samples of meat and meat products, 62 had concentrations above the LOQ values. The highest number of samples exceeding the LOQs were beef meat (11), pâté (9), and pork meat (7). The highest frequency quantification was determined for OCP compounds, with a total of 25 results (40.3%), of which 40% were DDT isomers (60% DDE). 

Among the individual compounds in fat samples (Table 2), hexachlorobenzene (HCB) and the isomer DDT, DDE were quantified in the highest percentages, HCB within the range of 5.5–66.7% and DDE in the range of 5.4–55.8%. HCBs were detected in the following order: game fat > sheep fat > goat fat > rabbit fat > beef fat > pork fat > poultry fat > horse fat > processed fat. Mean HCB concentrations found in the fat of animals and processed fat were in the range of 1.39–3.66 µg/kg. The detection frequency of DDE was determined in the following order: sheep fat > horse fat > game fat > goat fat > beef fat > pork fat > poultry fat > processed fat. The highest DDE concentration of 65.2 µg/kg was determined in horse fat. Mean values of the sum of DDT compounds (DDTs) ranged between 1.7 µg/kg in processed fat and 22.9 µg/kg in horse fat. The highest DDT concentration of 44.9 ug/kg was determined for poultry fat during 2017. In this study, HCHs were detected in only eight samples of beef fat and in only one sample each of pork, goat, sheep, and poultry fat. 

Despite being banned for a number of years and their disuse in agriculture, DDT and HCB are still found in the environment due to their persistence. When released into soil, HCB adsorbs strongly to organic matter and is generally considered immobile, with a half-life in soil estimated at 2.7–5.7 years in aerobic conditions and 10.6–22.9 years in conditions of anaerobic biodegradation [42]. DDT is persistent in the soil for up to 30 years, with a decline in DDT residues visible only in the third decade [43]. 

DDT can be microbially transformed under aerobic conditions in the soil into the stable and toxic metabolites DDE and DDD. The rate of transformation depends on factors such as soil type, organic carbon content, temperature, and moisture. The half-life of DDT in agricultural soils is 4–35 years (average 10–10.5 years) [44]. Studies on DDT distribution in animal tissues have shown that DDE is the principal contributor, whereas the contribution of DDD and DDT varies [45,46]. DDE is slowly eliminated and therefore tends to accumulate in animal tissues over time compared to other isomers. As such, it is the most stable DDT isomer [46,47]. This was confirmed in the present study, as DDE was quantified in the highest percentages in relation to other isomers of DDT, for example, in 95% of samples of beef fat. However, the measured DDT values were far below the MRL limits specified by the EU [28,46].

DDT can be found in different types of food, especially foods of animal origin [13]. In the framework of the total diet study (TDS) in Hong Kong, higher proportions of DDT and HCB were found in composite samples “meat, poultry and game and their products” (92% and 96%, respectively). The detectable levels of OCP residues were very low, up to 1.1 μg/kg [13]. Dietary exposure testing of DDT and HCH in food in Nanjing, China showed levels below the MRLs [6]. In different categories of food, the concentrations of DDT and HCH were determined in the following order: livestock meat > fish > pork > chicken. The highest concentration of DDTs and HCHs was detected in livestock meat (4.75 μg/kg). Also, the levels of DDT were higher than those of HCH in all food types. A recent study conducted in China showed DDTs of 2.7 μg/kg in meat samples [48]. Assessment of OCP residues on the market in Kenya showed the highest DDE mean concentration of 253.58 μg/kg in beef from Johannesburg [49]. A study in Spain found similar HCB concentrations (2.1 and 6.8 μg/kg) as the present study in conventionally and organically farmed beef [50]. Unlike in this study, HCBs were also determined in conventionally and organically farmed poultry meat in Spain. Sums of DDT in beef and poultry meat ranged from 14.9–25.6 μg/kg and 7.2–8.51 μg/kg, respectively.

In a study carried out in Bosnia and Herzegovina in traditionally and industrially smoked pork meat products, among 19 tested OCPs, only α-HCH and lindane were quantified above the LOQ, with the highest mean values of 15.05 μg/kg in cured pork neck after industrial smoking and 27.15 μg/kg in sausage ripening after industrial smoking [51]. High values of DDE and total DDT with maximal levels of 83 and 145 μg/kg were measured in pork fat tissues in Mexico [52].

Within the scope of the Multiannual National Control Programmes for Pesticides (EU MACP) in EU Member States during 2020, a total of 1595 samples of poultry fat were analysed. One sample exceeded the MRL (non-compliant), and two fat samples were found to contain DDT (one sample also contained HCB, the other dieldrin) [53]. Furthermore, in 2021, 1965 samples of bovine fat were analysed in EU Member States, of which 294 samples were reported as bovine meat, and 95.3% of samples were free of pesticide residues, whereas only one sample exceeded the MRL (non-compliant). Residues of DDT, beta-HCH, and lindane were reported in 0.97% (19) of those samples [54].

In this study, OPPs were determined in 25 samples of fat tissues and in four samples of meat and meat products. Among the fat samples, the most frequent compound exceeding the LOQ value (32%) was bromophos-ethyl. Chlorpyrifos was detected in five fat samples, with the highest concentration of 16.9 µg/kg in pork fat, which exceeded the prescribed MRL (10 µg/kg). For the group of pyrethroids and carbamates, 31 (only three were carbofuran) results exceeding the LOQ were found for fat tissues and 31 results for meat and meat product samples. Resmethrin concentrations above the MRL (20 µg/kg) were measured in six fat samples, ranging from 43.8 µg/kg in rabbit fat to 165.7 µg/kg in processed fat. Twelve positive results for pyrethroids were detected in pâtés, with the highest level of 25.9 µg/kg (>MRL) found for resmethrin. 

In a recent study in Korea, chlorpyrifos was determined in concentrations of 1.0 mg/kg in beef fat and 20 µg/kg in pork fat [55]. In a study from Brazil, chlorpyrifos was measured at a concentration of 42.1 µg/kg in beef fat, and cypermethrin was detected in all beef and poultry fat samples within the range of 1.44–5.85 and 0.59–6.25 µg/kg, respectively [21]. The highest level for pyrethroids was bifenthrin (16.9 µg/kg) measured in beef fat. 

Resmethrin is a synthetic pyrethroid insecticide that has been widely used since the 1970s to control insect pests in agriculture and public health [19]. It has insect control applications in residential, commercial, and industrial environments and in areas where animals live, and it is registered for use in food handling facilities and as a restricted use pesticide when used as a spray to control adult mosquitoes in the interest of public health. As liposoluble molecules, pyrethroids tend to migrate onto the lipid fraction of food, so the possibility of their bioaccumulation is higher in fatty types of food [21]. In assessment studies of population exposure to pesticides and pyrethroids through the consumption of different types of food in Brazil [18,21] and France [56,57], resmethrin was not quantified.

In the present study, the frequency quantification of PCB congeners in fat samples of beef, pork, goat, sheep, and poultry was (%) 19.3, 26,8, 15.8, 7.23, and 20.5, respectively. The highest frequency was determined in horse fat (36.7%). Congener PCB-153 was the most detected congener in beef fat (35.4%), pork fat (29.4%), and poultry fat (37.5%). The mean of PCBs ranged between 0.9 µg/kg in sheep fat and 5.27 µg/kg in poultry fat. The highest concentration was 32.6 µg/kg for PCB-28 in poultry fat. PCBs were determined in 12 meat and meat products and accounted for 24% of the detected results above the LOQs. Two samples of chicken meat were found to have the highest concentration of PCB-52, with a mean of 19.6 µg/kg.

In general, the obtained PCB levels in this study were lower than those presented in recent studies. Study of PCBs in meat and by-products from markets in Albania showed the highest PCBs of 21.88 µg/kg in poultry meat, followed by pork meat with 16 ug/kg [58]. A study from Bosnia and Herzegovina in traditionally and industrially smoked pork meat products detected PCB-28, PCB-52, and PCB-153, in which the highest mean values were determined in pancetta samples (µg/kg): 3.3 for PCB-28, 7.003 for PCB-52, and 8.003 for PCB-153 [51]. An evaluation of the mean values of PCBs obtained from the EU database for different countries showed PCB levels of 2.61 µg/kg in animal fat, 6.80 µg/kg in pork, and 12.7 µg/kg in poultry [2]. In a study conducted in southern Italy, significantly higher PCB concentrations were measured in meat and processed meat, with means of 39.7 and 20.2 µg/kg, respectively [59]. The lowest values of PCBs (0.97 µg/kg) were determined in meat samples in China [48].

As OCPs and PCBs are long-term soil pollutants, they can be introduced into the body of production animals, particularly cattle, pigs, sheep, and goats, via feed (grass, grass silage, or hay) or during grazing or free-range farming [60]. In the case of grazing, PCB is introduced via the soil, which depends on meadow quality and grass density. When grazing on dense meadows, a minimum intake of 3% soil is assumed, whereas grazing on low grass in the dry season or dirty grass in the wet season results in up to 10% of the soil being swallowed [61]. Given that PCB compounds are used in various products today and are therefore permanently present in the environment, some countries have set regulatory guidelines for their values in soils, depending on whether they are agricultural, industrial, or green areas. For example, in Germany, the highest values of 40 mg/kg were determined for industrial soils, and a value of 0.2 mg/kg was determined for lawns [62]. Furthermore, the Netherlands has set limit values for soils (0.08 mg/kg) in vegetable gardens in residential areas [63].

### 3.3. Dietary Exposure and Risk Characterisation Human Health Risk Assessment

Dietary exposure and potential chronic health exposure to pesticides and PCBs associated with consumption of fat of different animals, processed fat, and meat and meat products of the Croatian population are shown in Table 4. EDI values were calculated for quantified compounds for the most consumed types of fat, i.e., beef, pork, and poultry fat and processed fat. Also, risk assessments were conducted for beef, pork, and poultry meat for pesticide and PCB compounds that have been repeatedly quantified. 

EDI values varied for the quantified OCP (0.44 to 6.63 ng/kg bw/day) and OPP compounds (0.40 to 6.8 ng/kg bw/day). For DDTs, these values ranged from 0.7 to 6.68 ng/kg bw/day. For quantified pyrethroides and carbofuran, EDIs were within the range of 0.012–16.7 ng/kg bw/day. EDIs for resmethrin were between 22 and 66 ng/kg bw/day for fat of beef, pork, and poultry and processed fat. Daily intakes for PCBs were determined in the range of 0.03–2.10 ng/kg bw/day. 

Significantly higher daily intakes of beef and chicken meat for DDTs (10.71–18.37 ng/kg bw/day) and HCB (1.49–4.90 ng/kg bw/day) were determined in Spain [50]. In different studies, daily dietary exposures were calculated mainly on the basis of a total diet study, i.e., using different categories of food [6,13,48]. Dietary exposure to DDTs of 29.13 and 25.11 ng/kg bw/day for female and male adults, respectively, were determined in Nanjing, China [6]. Significantly higher EDI levels for PCBs and DDTs (17.8 and 75.2 ng/kg bw/day, respectively) were presented in a recent study from China [48]. The Total Dietary Study in France gave an EDI value for PCBs in meat of 0.178 ng/kg bw/day [64]. A study from Italy showed EDIs for PCBs from the consumption of meat and processed meat of 0.574 and 2.337 ng/kg bw/day, respectively [59].

In the present study, risk assessments for consumer health were carried out by calculating the HQ values, and the resulting EDIs were compared with the toxicological reference values defined at the level of European or international bodies and safety agencies. For all quantified compounds, the obtained HQ values were significantly lower than the toxicological reference values, i.e., significantly lower than 1, and for 80% of the calculated values, they were lower than 0.001. HQ values for PCBs ranged from 0.0015 to 0.11. Therefore, there is no potential health risk to consumers in relation to the pesticides included in this study through the consumption of these fat types. 

Some studies have shown HQs close to 1, such as for PCBs in China [6], and EDI values exceeding the toxicological limits, such as for carbofuran and diazinon [56]. A study in Spain showed risk regarding DDTs, HCHs, and HCB by the consumption of pork, chicken, and lamb meat with a calculated carcinogenic risk ranging between 1.76 and 17.41 for adults [50].

**Table 4 foods-13-00528-t004:** The estimation of daily intake and risk characterisation of pesticides and PCBs in beef, pork, and poultry fat, processed fat, beef, pork, and pâté.

Compound	TRV	Beef Fat		Pork Fat		Poultry Fat		Processed Fat		Beef		Pork		Pâté	
		EDI	HQ	EDI	HQ	EDI	HQ	EDI	HQ	EDI	HQ	EDI	HQ	EDI	HQ
Organochlorine pesticides (OCPs)															
Aldrin (aldrin + dieldrin)	100 ^b^			1.25	0.013	5.84	0.058					7.61	7.61 × 10^−5^		
Chlorobenzilate	20,000 ^a^	1.50	7.5 × 10^−5^	1.20	6.0 × 10^−5^	0.43	2.2 × 10^−5^	0.24	1.2 × 10^−5^			0.017	8.41 × 10^−7^		
∑ DDT	10,000 ^b^	1.30	1.3 × 10^−4^	0.90	9.0 × 10^−5^	1.90	1.9 × 10^−4^	0.70	7.0 × 10^−5^	6.63	6.63 × 10^−4^	0.014	1.38 × 10^−6^		
Endosulfan	6000 ^b^	0.94	1.5 × 10^−4^	0.44	7.3 × 10^−5^	0.47	7.8 × 10^−5^								
Endrin	200 ^b^	0.64	3.2 × 10^−3^	3.0	0.015										
∑ HCH	2000 ^d^	2.30	1.15 × 10^−3^	0.82	4.1 × 10^−4^	1.24	6.2 × 10^−4^								
Heptachlor	100 ^b^	2.40	0.024												
Hexachlorobenzene	170 ^c^	0.79	1.3 × 10^−4^	0.55	3.2 × 10^−3^	0.73	4.3 × 10^−3^	1.5	0.088	0.012	7.25 × 10^−5^				
Methoxychlor	100,000 ^a^	0.48	4.8 × 10^−6^	0.48	4.8 × 10^−6^										
Organophosphorus pesticides (OPPs)															
Bromophos-ethyl	3000 ^b^	0.48	1.6 × 10^−4^	0.52	1.7 × 10^−4^	0.40	1.3 × 10^−4^								
Chlorfenvinphos	500 ^b^	1.0	2.0 × 10^−3^												
Chlorpyrifos	10,000 ^b^	0.49	4.9 × 10^−5^	6.8	6.8 × 10^−4^										
Chlorpyrifos-methyl	10,000 ^b^	0.48	4.8 × 10^−5^	0.60	6.0 × 10^−5^										
Diazinon	200 ^a^	2.10	0.011	0.76	3.8 × 10^−3^										
Ethion	2000 ^b^			0.44	2.2 × 10^−4^										
Fenchlorphos	10,000 ^b^			0.52	5.2 × 10^−5^										
Pirimiphos-methyl	4000 ^a^	0.6	1.5 × 10^−4^												
Pyrethroids and carbamates (P and C)															
Bifenthrin *	15,000 ^a^	0.76	5.1 × 10^−5^	0.48	3.2 × 10^−5^										
Carbofuran	150 ^a^			5.30	0.035										
Cyfluthrin *	3000 ^a^			16.0	5.2 × 10^−5^										
Fenpropathrin	30,000 ^a^	1.5	5.0 × 10^−5^	1.8	6.0 × 10^−5^									0.017	3.86 × 10^−7^
Permethrin *	50,000 ^b^			2.1	4.2 × 10^−5^									0.012	2.34 × 10^−7^
Resmethrin *	30,000 ^a^	28.0	9.3 × 10^−4^	22.0	7.3 × 10^−4^	49.0	1.6 × 10^−3^	66.0	2.2 × 10^−3^					0.059	1.98 × 10^−6^
Polychlorobiphenyls (PCBs)															
∑PCB	20 ^d^	1.17	0.059	0.88	0.044	2.10	0.11	0.40	0.02			0.030	1.49 × 10^−3^		

TRV—Toxicological reference value (ng/kg bw/day). EDI—Estimated daily intake (ng/kg bw/day). HQ—Hazard Quotient. * Reported as the sum of isomers. ^a^ Acceptable daily intake (ADI) [39]. **^b^** Acceptable daily intake (ADI) and provisional tolerable daily intake (PTDI) [40]. ^c^ Tolerable daily intake (TDI) [65]. ^d^ Reference dose [41].

## 4. Conclusions

In summary, within the eight-year monitoring of the residues of 65 pesticides in fatty tissues of animals and meat and meat products, OCP compounds had the highest frequency of determination. In the fatty tissues of all types of animals, OCP compounds were measured in a frequency above 50%, i.e., above 73% for beef, sheep, goat, chicken, game, and processed fat samples. Furthermore, the obtained frequency quantifications of pesticides and PCBs in fatty tissues were significantly higher than those found in meat and meat products, which was to be expected considering the lipophilic properties of these compounds. The concentrations found in the investigated types of food are below the current levels defined by the EU MRL values, with the exception of resmethrin in six fat samples and clorpyrifos in one fat sample.

According to the quantified levels of pesticides and PCBs in fatty products and meat and meat products, a risk assessment was conducted to evaluate possible adverse impacts for consumer health. The risk assessment showed that it is unlikely that exposure to all 65 compounds analysed in this study would represent a detrimental health risk for consumers in Croatia.

The presented results are the first results summarising the exposure of adult consumers to pesticides and PCBs through the consumption of fat and meat and their products in Croatia. The determined results can also serve as a set of reference data for the exposure of children and teenagers, for which no data are currently available.

## Figures and Tables

**Table 1 foods-13-00528-t001:** Numbers of samples of fatty tissues, processed fat, and meat and meat products collected and analysed during the period 2015–2022.

Samples	Year of Collection								TOTAL
	2015	2016	2017	2018	2019	2020	2021	2022	
Fat and processed fat									
Pork fat	57	86	51	42	61	46	41	55	439
Beef fat	68	50	46	68	47	39	70	37	425
Poultry fat	21	21	40	19	17	33	23	20	194
Sheep fat	7	4	21	3	5	2	4	6	52
Horse fat	6	4	2	4	3	3	2	6	30
Goat fat	2	1	3	1	2	1	3	3	16
Game fat	1	2	1	1	1	1	1	1	9
Rabbit fat	1	1	1	1	1	1	1	1	8
Processed fat	7	9	11	15	22	19	14	14	111
TOTAL	170	178	176	154	159	145	159	143	1284
Meat and meat products									
Beef	21	20	14	12	13	16	10	8	106
Pork meat	41	37	31	68	25	29	21	15	267
Poultry meat	6	3	2	3	5	8	2	3	32
Processed meat	76	59	79	109	73	67	76	40	579
TOTAL	144	119	126	192	116	120	109	66	984

**Table 2 foods-13-00528-t002:** Concentrations of pesticides and PCBs in fat samples of different animals and processed fat sampled during the period 2015–2022.

Compound	Concentration ± SD (µg/kg) (Number of Results above the LOQ) Range (Min–Max)								
	Beef Fat N = 425	Pork Fat N = 439	Goat Fat N = 16	Sheep Fat N = 52	Poultry Fat N = 194	Game Fat N = 9	Horse Fat N = 30	Rabitt Fat N = 8	Processed Fat N = 111
Organochlorine pesticides (OCPs)									
Aldrin	<LOQ	<LOQ	<LOQ	<LOQ	14.6 ± 0.65 (2) 13.9–15.2	<LOQ	<LOQ	<LOQ	<LOD
Chlorobenzilate	3.80 ± 2.16 (6) 1.2–6.6	3.1 (1)	<LOQ	5.2 (1)	1.07 ± 0.04 (3) 1.0–1.1	<LOQ	<LOQ	<LOQ	5.6 (1)
DDE-p,p′	2.97 ± 2.43 (136) 1.0–19.4	2.0 ± 1.05 (24) 1.0–4.5	9.90 ± 13.2 (8) 0.5–43.1	4.50 ± 3.70 (26) 1.0–14.5	2.44 ± 1.29 (21) 1.1–5.3	5.60 ± 3.13 (6) 1.2–11.3	22.9 ± 29.9 (3) 2.0–65.2	3.37 ± 0.49 (3) 2.8–4.0	1.72 ± 0.65 (6) 1.1–3.0
DDD-p,p′	1.97 ± 0.78 (3) 1.0–2.9	1.7 (1)	<LOQ	<LOQ	2.0 (1)	1.6 (1)	<LOQ	<LOQ	<LOD
DDT-o,p′	1.85 ± 0.68 (12) 1.0–3.2	1.34 ± 0.24 (7) 1.0–1.8	<LOQ	<LOQ	2.10 ± 0.61 (4) 1.2–2.9	2.4 (1)	<LOQ	<LOQ	1.80 ± 0.99 (3) 1.0–3.2
DDT-p,p′	1.92 ± 1.36 (7) 1.0–2.6	1.71 ± 0.52 (7) 1.1–2.6	11.6 (1)	2.1 (1)	13.3 ± 16.6 (3) 1.1–36.8	1.5 (1)	<LOQ	<LOQ	<LOD
∑DDT	3.14 ± 2.76 (141) 1.0–19.4	2.25 ± 1.21 (32) 1.0–4.5	11.4 ± 12.8 (8) 0.5–43.1	4.58 ± 3.71 (26) 1.0–14.5	4.67 ± 9.15 (22) 1.1–44.9	6.52 ± 3.84 (6) 1.2–11.3	22.9 ± 29.9 (3) 2.0–65.2	3.37 ± 0.49 (3) 2.8–4.0	1.70 ± 0.76 (6) 1.0–3.2
Dieldrin	<LOQ	3.13 ± 0.81 (6) 1.5–4.0	<LOQ	3.87 ± 0.21 (3) 3.6–4.1	2.65 ± 1.35 (2) 1.3–4.0	<LOQ	<LOQ	<LOQ	<LOD
Endosulfan	2.35 ± 0.05 (2) 2.3–2.4	1.1 (1)	<LOQ	<LOQ	1.17 ± 0.05 (3) 1.1–1.2	<LOQ	<LOQ	<LOQ	<LOD
Endrin	1.6 (1)	7.5 (1)	<LOQ	<LOQ	<LOQ	<LOQ	<LOQ	<LOQ	<LOD
HCH, alpha-	1.53± 0.37 (3) 1.1–2.0	<LOQ	<LOQ	<LOQ	<LOQ	<LOQ	<LOQ	<LOQ	<LOD
HCH, beta-	3.05 ± 0.45 (2) 2.6–3.5	<LOQ	2.2 (1)	<LOQ	<LOQ	<LOQ	<LOQ	<LOQ	<LOD
HCH, gamma-/Lindan	1.17 ± 0.37 (3) 0.7–1.6	2.05 ± 0.85 (2) 1.2–2.9	<LOQ	1.5 (1)	3.10 ± 1.40 (2) 1.7–4.5	<LOQ	<LOQ	<LOQ	<LOD
Heptachlor	5.9 (1)	<LOQ	<LOQ	<LOQ	<LOQ	<LOQ	<LOQ	<LOQ	<LOD
Hexachlorobenzene	1.99 ± 1.11 (139) 1.0–8.25	1.39 ± 0.23 (20) 1.0–1.9	3.48 ±2.00 (6) 1.3–7.2	3.47 ± 2.43 (29) 1.2–10.7	1.83 ± 1.52 (16) 0.8–6.2	2.80 ± 1.06 (4) 1.0–3.6	1.89 ± 0.80 (15) 1.1–4.3	<LOQ	3.66 ± 3.90 (5) 1.5–11.4
Methoxychlor	1.2 (1)	1.2 (1)	<LOQ	<LOQ	<LOQ	<LOQ	<LOQ	<LOQ	<LOD
Organophosphorus pesticides (OPPs)									
Bromophos-ethyl	1.2 (1)	1.30 ± 0.29 (5) 1.0–1.7	<LOQ	1.8 (1)	1.0 (1)	<LOQ	<LOQ	<LOQ	<LOD
Chlorfenvinphos	2.60 ± 1.40 (2) 1.0–1.5	<LOQ	<LOQ	<LOQ	<LOQ	<LOQ	<LOQ	<LOQ	<LOD
Chlorpyrifos	1.23 ± 0.20 (3) 1.2–4.0	16.9 (1)	<LOQ	<LOQ	1.2 (1)	<LOQ	<LOQ	<LOQ	<LOD
Chlorpyrifos-methyl	1.2 (1)	1.5 (1)	<LOQ	<LOQ	<LOQ	<LOQ	<LOQ	<LOQ	<LOD
Diazinon	5.30 ± 2.9 (2) 2.4–8.2	1.90 ± 0.2 (2) 1.7–2.1	<LOQ	14.2 (1)	<LOQ	<LOQ	<LOQ	<LOQ	<LOD
Ethion	<LOQ	1.1 (1)	<LOQ	<LOQ	<LOQ	<LOQ	<LOQ	<LOQ	<LOD
Fenchlorphos	<LOQ	1.3 (1)	<LOQ	<LOQ	<LOQ	<LOQ	<LOQ	<LOQ	<LOD
Pirimiphos-methyl	1.5 (1)	<LOQ	<LOQ	<LOQ	<LOQ	<LOQ	<LOQ	<LOQ	<LOD
Pyrethroids and carbamates (P and C)									
Allethrin	41.9 ± 27.6 (5) 7.4–74.1	<LOQ	<LOQ	<LOQ	39.1 ± 21.9 (2) 17.2–60.9	<LOQ	1.9 (1)	<LOQ	<LOD
Bifenthrin *	1.9 (1)	1.2 (1)	<LOQ	<LOQ	<LOQ	<LOQ	<LOQ	<LOQ	<LOD
Carbofuran	<LOQ	13.3 (1)	6.3 (1)	3.68 (1)	<LOQ	<LOQ	<LOQ	<LOQ	<LOD
Cyfluthrin *	<LOQ	39.3 (1)	<LOQ	<LOQ	<LOQ	<LOQ	<LOQ	<LOQ	<LOD
Fenpropathrin	3.67 ± 2.32 (7) 1.9–9.1	4.43 ± 3.02 (3) 2.2–8.7	<LOQ	<LOQ	<LOQ	<LOQ	<LOQ	<LOQ	<LOD
Permethrin *	<LOQ	5.25 ± 1.45 (2) 3.8–6.7	<LOQ	<LOQ	<LOQ	<LOQ	<LOQ	<LOQ	<LOD
Resmethrin *	70.8 ± 5.15 (2) 65.6–75.9	54.3 (1)	<LOQ	<LOQ	121.9 (1)	<LOQ	<LOQ	43.8 (1)	165.7 (1)
Polychlorobiphenyls (PCBs)									
PCB-28	0.93 ± 0.37 (8) 0.5–1.7	0.57 ± 0.09 (3) 0.5–0.7	1.0 (1)	<LOQ	15.8 ± 14.9 (4) 1.0–32.6	<LOQ	<LOQ	<LOQ	0.7 (1)
PCB-52	1.09 ± 0.83 (11) 0.5–2.7	0.73 ± 0.14 (6) 0.5–0.9	<LOQ	<LOQ	0.5 (1)	<LOQ	<LOQ	1.2 (1)	<LOD
PCB-101	1.91 ± 1.06 (7) 0.6–3.8	0.83 ± 0.15 (3) 0.6–1.0	<LOQ	<LOQ	1.0 (1)	1.2 (1)	<LOQ	<LOQ	<LOD
PCB-118	2.29 ± 1.06 (9) 0.8–10.9	1.00 ± 0.37 (3) 0.5–1.4	<LOQ	<LOQ	<LOQ	<LOQ	<LOQ	<LOQ	<LOD
PCB-138	1.78 ± 0.49 (11) 1.3–2.3	2.50 ± 1.74 (5) 0.8–5.7	<LOQ	<LOQ	1.63 ± 0.80 (4) 0.8–2.9	<LOQ	2.35 ± 1.31 (5) 1.2–4.5	<LOQ	1.15 ± 0.15 (2) 1.0–1.3
PCB-153	1.34 ± 0.89 (29) 0.5–3.7	1.59 ± 1.33 (10) 0.6–4.9	0.9 (1)	0.90 ± 0.30 (5) 0.6–1.4	1.38 ± 0.81 (6) 0.5–2.9	0.9 (1)	1.0 ± 0.46 (5) 0.6–1.7	<LOQ	1.4 (1)
PCB-180	2.61 ± 1.25 (7) 1.1–4.6	1.63 ± 1.10 (4) 0.5–3.4	1.5 (1)	<LOQ	<LOQ	<LOQ	1.5 (1)	<LOQ	<LOD
∑PCB	2.93 ± 4.09 (47) 0.5–26.2	2.20 ± 3.07 (23) 0.5–14.0	3.4 (1) 0.9–1.5	0.90 ± 0.30 (5) 0.6–1.4	5.27 ± 9.79 (16) 0.5–32.6	2.1 (1) 0.9–1.2	2.27 ± 1.43 (7) 0.6–4.5	<LOQ	1.0 ± 0.24 (3) 0.7–1.3
Total no of detected results	424	127	19	69	78	16	30	5	19
Total no of OCP	317	73	16	61	57	14	18	3	15
Total no of OPP	10	11		2	2				
Total no of P and C	15	9	1	1	3		1	1	1
Total no of PCBs	82	34	3	5	16	2	11	1	4

* Reported as the sum of isomers.

**Table 3 foods-13-00528-t003:** Concentrations of detected pesticides and PCBs above LOQ in meat and meat products in the period 2015–2022.

Meat and Meat Products	Detected Pesticide or PCBs	Number of Results above the LOQ	Concentration ±SD (µg/kg)	Range (µg/kg)
Beef	Aldrin	1	2.80	
	DDT-p,p′	1	1.1	
	DDE-p,p′	3	2.50 ± 2.19	1.1–6.3
	Hexachlorobenzene	2	4.65 ± 2.05	1.6–6.7
	Allethrin	5	11.4 ± 3.92	2.8–16.4
	PCB 28	1	4.80	
	PCB 101	1	10.2	
Beef breakfast meat	DDE-p,p′	1	1.4	
	Chlorpyrifos-methyl	1	1.5	
	PCB 28	1	4.8	
Pork meat	Aldrin	3	2.87 ± 0.31	2.6–3.3
	DDT-p,p′	1	2.1	
	DDD-p,p′	2	5.20 ± 0.10	5.1–5.3
	Chlorobenzilate	2	6.35 ± 0.05	6.3–6.4
	Hexachlorobenzene	1	1.6	
	PCB 28	3	5.10 ± 0.37	4.6–5.5
	PCB 118	2	6.15 ± 0.05	6.1–6.2
Pork vegetable stew	Cypermethrin *	1	17.8	
Poultry meat	Parathion-ethyl	1	1.4	
	PCB 52	2	19.6 ± 1.15	18.4–20.7
Pâté	DDE-p,p′	1	1.4	
	Fenpropathrin	3	4.37 ± 1.45	1.8–6.6
	Permethrin *	7	4.43 ± 2.98	1.2–10.8
	Resmethrin *	2	22.4 ± 3.50	18.9–25.9
	PCB 28	1	0.53	
Sausage	Pentachloroaniline	1	7.6	
	Pirimiphos-methyl	1	1.0	
	Permethrin *	1	5.9	
Ham	Chlordane, cis-	1	2.3	
	DDE-p,p′	2	1.1 ± 0.1	1.1–1.3
	DDD-p,p′	2	3.2 ± 2.1	1.1–5.3
	Fenthion	1	1.3	
	Allethrin	1	3.9	
	Resmethrin *	1	13.1	
	PCB 118	1	6.4	
Mortadella salami	DDE-p,p′	1	1.5	

* Reported as the sum of isomers.

## Data Availability

Data is contained within the article or Appendix A.

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
