# Peer review of "Distribution of Pesticides and Polychlorinated Biphenyls in Food of Animal Origin in Croatia"

_foods, 2024, doi:10.3390/foods13040528_

Round 1

Reviewer 1 Report

Comments and Suggestions for Authors

The manuscript has important information and results for the literature.   It can be evaluated as original. The sections were suitable for the journal rules. The references are appropriate. The manuscript has different information about the subject and it has a suitable methodology. The tables were suitable but the figures can be formed. This paper provides new results in related fields. On the other hand, the following factors should be under consideration;

·         A graphical abstract can be formed.

·         The general situation (purposes, harmful of pesticides, etc) can be written into the abstract.

·         At line 101, the term ‘(EC, 2006, 2023b)’ was necessary or not.

·         The word of rabbit should be changed to ‘rabitt ’at table 1

·         What is the reason of the samples of sheep fat, horse fat, goat fat, game fat, and rabbit fat were lower than pork, beef and poultry? Can the mentioned samples be used for general evaluation?

Reviewer 2 Report

Comments and Suggestions for Authors

The manuscript documents a comprehensive survey regarding the occurrence OPPs, OCPs and PCBs in meat-based food products, featuring the analysis of >2,000 samples and >70 analytes. Furthermore, the authors also estimate dietary exposure based on the survey results. In general, the reported data are supported by good QC practice. Below please find a few minor comments for consideration.

1. Specify what was the internal standard used for the study.

2. How come the internal standard was not spiked into the sample prior to extraction?

3. For pyrethroids, clarify how concentrations were calculated. Reported as the sum of the individual isomers?

4. Line 53: replace "chemical and physical" with "physicochemical"

5. Line 374: missing the reference regarding the study Bosnia and Herzegovina.

6. Curiosity. How come no dioxin-like PCBs were included in the study. 
